# Generative Recorrupted-to-Recorrupted: An Unsupervised Image Denoising Network for Arbitrary Noise Distribution

## Abstract

With the great breakthrough of supervised learning in the field of denoising, more and more works focus on end-to-end learning to train denoisers. The premise of this method is effective is that there is certain data support, but in practice, it is particularly difficult to obtain labels in the training data. To this end, some unsupervised denoisers have emerged in recent years, however, the premise of these methods being effective is that the noise model needs to be known in advance, which will limit the practical use of unsupervised denoising. In addition, inaccurate noise prior from noise estimation algorithms causes low denoising accuracy. Therefore, we design a more practical denoiser that requires neither clean images as training labels nor noise model assumptions. Our method also needs the support of the noise model, the difference is that the model is generated by a residual image and a random mask during the network training process, and then the input and target of the network are generated from a single noisy images and the noise model, at the same time, train an unsupervised module and a pseudo supervised module. Extensive experiments demonstrate the effectiveness of our framework and even surpass the accuracy of supervised denoising.

## 1 Introduction

Image denoising is a traditional topic in the field of image processing, and it is the basis for the success of other vision tasks. A noisy image can be represented by $y = x + n$, and our task is to design a denoiser to remove the noise in the noisy image.

The denoising convolutional neural network DNCNN Zhang et al. (2017) can be considered a benchmark of the use of deep learning for image denoising, and it introduced residual learning and batch normalization, which speed up the training process as well as boost the denoising performance. The fast and lexible denoising neural network FFDNET Zhang K. & Zhang (2018) treated the noisy model as a prior probability distribution, such that it can effectively handles wide a range of noise levels The convolutional blind denoising CBDNET Guo et al. (2019) went further than the FFDNET Zhang K. & Zhang (2018) and aimed at real photographs though synthesized and real images were both used in training.

A common treatment for the above methods is that it all need to take noisy-clean image pairs in training. However, in some scenarios such as medical and biological imaging, there often lack clean images, leading to an infeasibility of the above methods. To this end, the noise-to-noise(N2N) method Lehtinen et al. (2018) was the first to reveal that deep neural networks (DNNs) can be trained with pairs of noisy-noisy images instead of noisy-clean images, in other words, training can be conducted with only two noisy images that are captured independently in the same scene. The N2N can be used in many taskBuchholz et al. (2019); Ehret et al. (2019); Hariharan et al. (2019); Wu et al. (2019); Zhang et al. (2019), since it creatively addressed the dependency on clean images. Unfortunately, pairs of corrupted images are still difficult to be obtained in dynamic scenes with deformations and image quality variations.

To further bring the N2N into practice, some research Krull et al. (2019); Batson & Royer (2019); Krull et al. (2020); Huang et al. (2021) concluded that it is still possible to train the network without using clean images if the noise between each region of the image is independent. Among them,

Neighbor2Neighbor (NBR2NBR) method Huang et al. (2021) proposed a new sampling scheme to achieve better denoising effects with a single noisy image. The advantage of this approach is that it does not need a prior noise model prior, like the Recorrupted-to-Recorrupted (R2R) Pang et al. (2021), nor does it lose image information, like the Noise2Void (N2V) self-supervised methods N2V Krull et al. (2019).

However, Krull et al. (2019); Batson & Royer (2019); Krull et al. (2020); Huang et al. (2021); Pang et al. (2021) are valid under the assumption that the noise on each pixel is independent from each other, which means that they are not as effective in dealing with noise in real scenes as supervised denoising. To deal with more complex noise, some unsupervised methods have been proposed. Noise2Grad (N2G) Lin et al. (2021) extracted noise by exploiting the similar properties of noise gradients and noisy image gradients, and then added the noise to unpaired clean images to form paired training data. Wang et al. (2022a) constructed a new way of unsupervised denoising through optimal transport theory. It is worth noting that although Lin et al. (2021); Wang et al. (2022a) no longer subject the end-to-end learning approach to pairing clean-noisy images, they still need to collect many clean images.

In order to solve the above problems, we propose a new denoising mode that achieves unsupervised denoising without requiring noise prior, noise assumptions and any clean images. In one epoch of our training network, we first obtain a residual image containing noise through the difference between network input and output, and use a random mask image to reduce the influence of natural image information in the residual image. The generative noise model can be obtained by the above operation. In the second step, we put the model into the Pseudo Supervised module and the Recorrupted-to-Recorrupted module to train the same network. At this point, one epoch of the entire training ends. Eventually after many iterations of network training, we will gradually generate more realistic noise model and perfect denoiser. More detail of our proposed Generate Recorrupted-to-Recorrupted framework can be found in Figure 1.

The remainder of the paper is organized as follows. In Section 2, we introduce the related work. Then the details of our method is given in Section 3, followed by the experiments in Section 4, with the conclusions drawn in Section 5.

## 2 PROBLEM STATEMENT

### 2.1 SUPERVISED TRAINING

With the rapid development of deep learning, many supervised learning methods are applied to image denoising. DNCNN Zhang et al. (2017) successfully applied a 17-layer deep neural network to image denoising tasks by introducing residual learning. After that, a series of more efficient and complex neural networks succeed in denoising tasks. Unlike DNCNN Zhang et al. (2017), FFDNET Zhang K. & Zhang (2018) was more efficient in denoising, and CBDNET Guo et al. (2019) can handle more complex real noise. Without considering constraints, the above supervised denoising methods can be expressed by the following formula:

$$\operatorname*{argmin}_{\theta} L\left(f_\theta\left(y\right), x\right). \tag{1}$$

$y$, $x$, $f$ and $L$ are noisy images, clean images, denoising model and loss function respectively. However, these methods all require clean images as the target of training the neural network, and then optimize the parameter $\theta$ by calculating the gap between the network output and the target, so as to obtain a better denoising model.

The N2N Lehtinen et al. (2018) revealed that the noisy/true image pairs used to train the DNN can be replaced by noisy/noisy images pairs. The corrupted pairs are represented by the $y$ and $z$, where $n_1$ and $n_2$ are uncorrelated. There are two main principles for N2N to successfully train network with a paired noisy image: the first is that the optimal solution obtained by network training is a mean value solution; the second is that the mean value of most noise is close to zero. So the gradient generated by the network for one corrupted target is incorrect, but the gradient corresponding to the average of all corrupted images is correct, which can be expressed by the following formula:

$$\begin{aligned} &\operatorname*{argmin}_{\theta} L\left(f_\theta\left(y\right), z\right) \\ &y = x + n_1 \\ &z = x + n_2. \end{aligned} \tag{2}$$

Although the N2N alleviated the dependence on clean images, pairs of noisy images are still difficult to obtain.

## 2.2 SELF-SUPERVISED TRAINING

To eliminate N2N Lehtinen et al. (2018) restrictions in dynamic scene denoising, some methods utilized the original features of noisy images to construct auxiliary tasks to make self-supervision as effective as supervision. The N2V Krull et al. (2019) offered a blind-spot network which can denoise using only a noisy image, and its main principle is to use the correlation among pixels to predict missing pixels. In this way, some pixel information is lost and the denoising effect is not ideal. The Self2Self Quan et al. (2020) can realize image denoising by dropout in the case of only one noisy image. Unlike N2V, it used lost information as targets, but its training time was extremely long and test a image needed a training model, which limit its practical application.

The NBR2NBR Huang et al. (2021) was the most recent a self-supervised denoising method, which can be viewed as an advanced version of the N2N Lehtinen et al. (2018) via a novel image sampling scheme to get rid of the requirement of paired noisy images.

The following equation shows the reason why denoising is effective without clean targets under the condition that the noise mean is 0:

$$\mathbb{E}_{x,y}\|f_\theta(y) - x\|_2^2 = \mathbb{E}_{a,y}\|f_\theta(y) - a\|_2^2 + const \\ + Cov((f_\theta(y) - x), (a - x)). \tag{3}$$

In N2N Lehtinen et al. (2018), $a = z$, since $n_1$ and $n_2$ are independent covariance terms will disappear. Similarly in NBR2NBR Huang et al. (2021), a and y are replaced by two adjacent noisy sub-images, covariance terms disappears under the local correlation of pixels and assumption of uncorrelated noise. Also in N2V Krull et al. (2019), it is denoised by a blind spot network, which removes the covariance term by assuming that the noise is uncorrelated on adjacent pixels.

## 2.3 UNSUPERVISED TRAINING

A category of unsupervised denoising requires unpaired clean-noisy images to train the denoiser. N2G Lin et al. (2021) obtained an approximate noise model by measuring the distance between the noise gradient generated after denoising and the original noisy image gradient, and then added it to the unpaired clean images for training. However, the approach requires not only clean images but also a lot of time to approximate the real noise distribution. Another approach Wang et al. (2022a) utilized WGAN-gp Gulrajani et al. (2017) to measure the distance between the denoised image output by the generator and the unpaired clean image to optimize the denoiser.

Another category of unsupervised denoising is when certain assumptions about the noisy model are required. AmbientGAN Bora et al. (2018), a method of training generative adversarial networks also disposed the reliance on clean images, and used a measurement function to generate a noisy image, which was then fed into the discriminator for comparison with the original noisy image. The Noiser2Noise (Nr2N) Moran et al. (2020) re-destroyed the original noisy images through the noise prior as the input $\hat{y} = y + \hat{n}$ to the network, and the original noisy images $y$ were used as training targets. Then, R2R Pang et al. (2021) overcame the disadvantage of N2N Lehtinen et al. (2018) and generated paired noisy images by introducing a prior noise model. In particular, it used known noise levels to form pairs of noisy images for a single noisy image. Although R2R is close to N2N Lehtinen et al. (2018) in denoising effect, it is limited in applications with real noisy images because the level and type of noise from real noisy images are difficult to measure.

## 3 THEORETICAL FRAMEWORK

Our framework consists of three modules: Generate Noise (GN), Pseudo Supervised (PS) and Recorrupted2Recorrupted (R2R). These three modules share a neural network parameter during training, which means they are trained simultaneously, but in a different order in a network training iteration. First, we generate a simulated noise through GN. In the second step, the noise is put into PS and R2R, and the network parameter is updated at the same time. In the next round of iterative training, GN will use the new network parameter to generate more realistic noise.

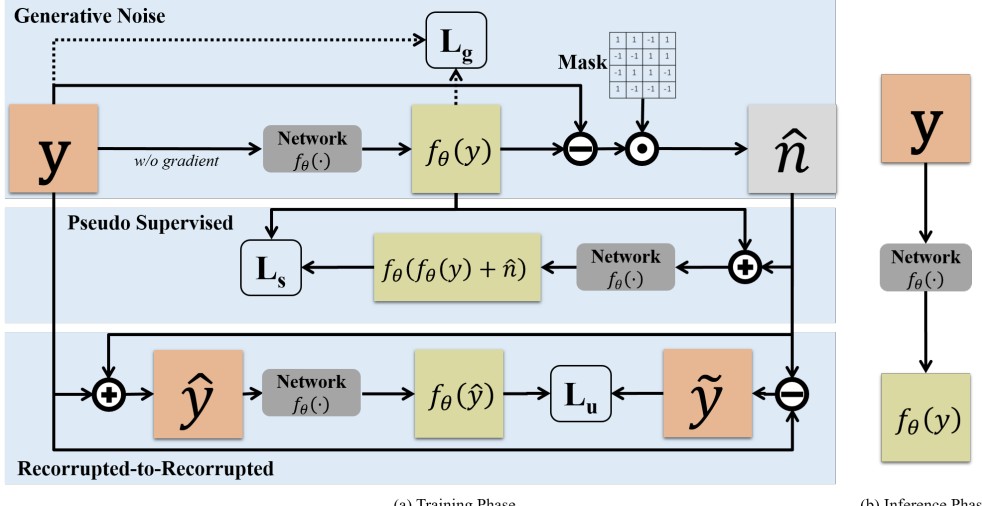

(a) Training Phase           (b) Inference Phase

Figure 1: The framework of Generate Recorrupted-to-Recorrupted (GR2R). (a) Overall training process. $y$ is the observed noisy image, the generate noise $\hat{n}$ in Pseudo Supervised (PS) and Recorrupted2Recorrupted (R2R) is obtained by element-wise multiplication of random mask $m$ and residual map $y - f_\theta(y)$. The neural networks in the three modules update the same parameter $\theta$ in one network. Moreover, the regular loss $L_g$ is used to stabilize the training phase, the supervised loss $L_s$ avoids randomly generated noise from affecting unsupervised denoising, the $L_u$ represents unsupervised loss. (b) Inference using the trained denoising model. The denoising network generates denoised images directly from the noisy images $y$ of the test set without additional operations.

## 3.1 RECORRUPTED2RECORRUPTED MODULE

R2R Pang et al. (2021) is an unsupervised denoising method using a training scheme that does not require noisy image pairs or clean target images. The approach destroys a single noisy image through the noise prior to form the paired noisy images required in N2N Lehtinen et al. (2018). Given the noisy observation $y$ and noise prior $n$, R2R aims to minimize the following empirical risk:

$$\operatorname*{argmin}_{\theta} L\left(f_\theta\left(\hat{y}\right), \tilde{y}\right)$$
$$\hat{y} = y + n'$$
$$\tilde{y} = y - n'. \tag{4}$$

where $f_\theta\left(\bullet\right)$ denotes the denoising model with parameter $\theta$. This method assumes that noise $n'$ is known before training network, however, it is difficult to accurately estimate the noise model in real noisy image denoising, which affects the performance of the denoiser. The loss of equation 4 is closely related to the one used in supervised learning:

$$\operatorname*{argmin}_{\theta} L\left(f_\theta\left(\hat{y}\right), x\right). \tag{5}$$

Proofs are as follows:

$$\mathbb{E}_{n,n'}\left\{\|f_\theta(\hat{y}) - \tilde{y}\|_2^2\right\} = \mathbb{E}_{n,n'}\left\{\|f_\theta(\hat{y}) - x - n + n'\|_2^2\right\}$$
$$= \mathbb{E}_{n,n'}\left\{\|f_\theta(\hat{y}) - x\|_2^2\right\}$$
$$- 2\mathbb{E}_{n,n'}\left\{(n - n')^\top\left(f_\theta(\hat{y}) - x\right)\right\} \tag{6}$$
$$+ \mathbb{E}_{n,n'}\left\{(n - n')^\top(n - n')\right\} \tag{7}$$
$$= \mathbb{E}_{n,n'}\left\{\|f_\theta(\hat{y}) - x\|_2^2\right\}.$$

The $n$, $n'$, $x$ are the noise in the observed image, the noise prior and the clean image, respectively. In equation 6, as long as $n$ and $n'$ are independent, it can be inferred that $n + n'$ in $f_\theta(\hat{y})$ and $n - n'$

are independent. According to the assumption of N2N Lehtinen et al. (2018) that the noise mean is 0, it can be found that equation 6 is equal to 0. In equation 7, since the variance of the noise prior $n'$ is equal to the variance of the noise $n$, equation 7 is exactly equal to 0. Next, going a step further, how to achieve unsupervised denoising without noise prior?

## 3.2 Generate Noise Module

A natural idea is to generate the noisy model at training time. First we generate a residual image by following equation $y - f_\theta(y)$, which contains real noise. However, since the residual image also contains a large amount of image feature information, it is difficult to accurately model noise. So in the second step we introduce a random mask map $m$, which is a vector of the same size and dimension as the noisy image, and it obeys the following distribution:

$$m = \begin{cases} 1 & p = 0.5 \\ -1 & p = 0.5. \end{cases} \tag{8}$$

Then the generated noise during training can be represented as follows:

$$\hat{n} = (y - f_\theta(y)) \odot m. \tag{9}$$

where $\odot$ represents element-wise multiplication. It can be seen from equation 8 that $m$ is randomly generated and has a mean value of 0. From equation 9, it can be deduced that the mean value of $\hat{n}$ is 0. Even if the original real noise $n$ does not satisfy the assumption that the noise from Lehtinen et al. (2018) has a mean of 0, the generated noise can be forced to satisfy the assumption of zero mean during network training iterations by equation 9. In addition, the operation may produce a slight error in estimating the real noise model, so we design the Pseudo Supervised module to reduce this error.

The input and target of our Recorrupted2Recorrupted are represented by the following formulas:

$$\begin{aligned} \hat{y} &= y + \hat{n} \\ \tilde{y} &= y - \hat{n}. \end{aligned} \tag{10}$$

It is easy to see that the generated $\hat{n}$ and $n$ are irrelevant, so $n + \hat{n}$ and $n - \hat{n}$ are unrelated. At this time, under the generated noise $\hat{n}$ scheme, equation 6 vanish as in Recorrupted2Recorrupted. Since the variance of $m$ is 1, equation 7 can be transformed into the following formula:

$$\mathbb{E}_{n,\hat{n}} \left\{ \|f_\theta(y) - y\|_2^2 \right\} + const. \tag{11}$$

Therefore, an optimal denoising criterion to achieve the same effect of supervised denoising can be expressed as

$$\underset{\theta}{\text{argmin}} L\left(f_\theta\left(\hat{y}\right), \tilde{y}\right) + \underset{\theta}{\text{argmin}} L\left(f_\theta\left(y\right), y\right). \tag{12}$$

## 3.3 Pseudo Supervised

Note that both equation 4 and equation 12 are equivalent to supervised denoising equation 5. However, in equation 5, the extra noise introduced by $\hat{y}$ will affect the accuracy in the denoiser, so R2R Pang et al. (2021) adopts the Monte Carlo approximation to solve above problem. However, the averaging of multiple forward processes in R2R will not only greatly reduce the denoising speed on the test set but also affect the denoising accuracy. In addition, the noise we generate will have a certain error, so we design a supervised-like approach to address the effect of the generate noise. Specifically, we use the generated $f_\theta(y)$ in GN as the 'clean' target and $f_\theta(y) + \hat{n}$ as the input to train the denoiser. During training, since GN stop the update of $\theta$, $f_\theta(y)$ gradually approaches the clean image without affecting the stability of training. The pseudo supervised loss as follows:

$$\underset{\theta}{\text{argmin}} L\left(f_\theta\left(f_\theta\left(y\right) + \hat{n}\right), f_\theta\left(y\right)\right). \tag{13}$$

The total loss function of our method can be expressed as:

$$\begin{aligned} L &= L_u + L_g + \gamma \cdot L_s \\ &= \|f_\theta\left(\hat{y}\right) - \tilde{y}\|_2^2 + \|f_\theta\left(y\right) - y\|_2^2 + \gamma \cdot \|f_\theta\left(f_\theta\left(y\right) + \hat{n}\right) - f_\theta\left(y\right)\|_2^2 \end{aligned} \tag{14}$$

where $f_\theta$ a denoising network with arbitrary network design, and $\gamma$ is a hyper-parameter controlling the strength of the pseudo supervised term.

## 4 EXPERIMENTS

In this section, we evaluate our GR2R framework, which can bring significant improvement to the denoising quality of previous work.

**Training Details.** We use the same modified U-Net Ronneberger et al. (2015) architecture as Lehtinen et al. (2018); Huang et al. (2021); Wang et al. (2022b). The batch size is 10. We use Adam Kingma & Ba (2015) as our optimizer. The initial learning rate is 0.0003 for synthetic denoising in sRGB space and 0.0001 for real-world denoising. All models are trained on a server using Python 3.8.5, Pytorch 1.6, and Nvidia Tesla K80 GPUs.

**Datasets for Synthetic Denoising.** Following the setting in Lehtinen et al. (2018); Huang et al. (2021), we select 44,328 images with sizes between $256 \times 256$ and $512 \times 512$ pixels from the ILSVRC2012 Deng et al. (2009) validation set as the training set. To obtain reliable average PSNRs, we also repeat the test sets Kodak Franzen (1999), BSD300 Martin et al. (2001) and Set14 Zeyde et al. (2010) by 10, 3 and 20 times, respectively. Thus, all methods are evaluated with 240, 300, and 280 individual synthetic noise images. Specifically, we consider two types of noise in sRGB space: (1) Gaussian noise with $\sigma = 25$ , (2) Gaussian noise with $\sigma \in [5, 50]$.

**Datasets for Real-World Denoising** Following the setting in Huang et al. (2021), we take the SIDD Abdelhamed et al. (2018) dataset collected by five smartphone cameras in 10 scenes under different lighting conditions for real-world denoising in raw-RGB space. We use only raw-RGB images in SIDD Medium Dataset for training and use SIDD Validation and Benchmark Datasets for validation and testing.

**Details of Experiments.** For the baseline, we consider two supervised denoising methods (N2C Ronneberger et al. (2015) and N2N Lehtinen et al. (2018)). We also GR2R with a traditional approach (BM3D Dabov et al. (2007)) and eight self-supervised denoising algorithms (Self2Self Quan et al. (2020), Noise2Void (N2V) Krull et al. (2019), Laine19 Laine et al. (2019), Noisier2Noise Moran et al. (2020), DBSN Wu et al. (2020), R2R Pang et al. (2021), NBR2NBR Huang et al. (2021) and B2UB Wang et al. (2022b)).

### 4.1 RESULTS FOR SYNTHETIC DENOISING

The quantitative comparison results of synthetic denoising for gaussian can be seen in Table 1. Whether the gaussian noise level is fixed or variable, our approach significantly outperforms the traditional denoising method BM3D and all self-supervised denoising methods on BSD300 dataset, even beyond supervised learning methods. On the other two small test sets (KODAK and SET14), our method is also close to Laine19-pme Laine et al. (2019), which is a method that requires the same explicit noise modeling as R2R Pang et al. (2021). However, explicit noise modeling means strong prior, leading to poor performance on real data. The following experiments on real-world datasets also illustrate this problem. In addition, figure 2 shows that our method retains more natural image features while denoising compared to other denoisers.

### 4.2 RESULTS FOR REAL-WORLD DENOISING

In real raw-RGB space, Table 2 shows the quality scores for quantitative comparisons on SIDD Benchmark and SIDD Validation. Note that the online website of SIDD evaluates the quality scores for the SIDD Benchmark. Surprisingly, the proposed method outperforms the state-of-the-art (NBR2NBR) by 0.28 dB and 0.23 dB for the benchmark and validation, but even outperforms N2C and N2N about 0.1 dB. It is worth noting that the unsupervised methods Laine et al. (2019) and Pang et al. (2021) relying on model prior are significantly less effective in dealing with real noise, and we even surpass Pang et al. (2021) by 4.05 dB and 4.09 dB for the benchmark and validation. Obviously this type of model prior based approach is not advisable. The raw-RGB denoising performance in the real world demonstrates that our method is able to simulate complex real noise distributions.

### 4.3 ABLATION STUDY

This section conducts ablation studies on the pseudo supervised module. Table 3 lists the performance of different $\gamma$ values in unsupervised denoising. It can be seen that the denoising effect is

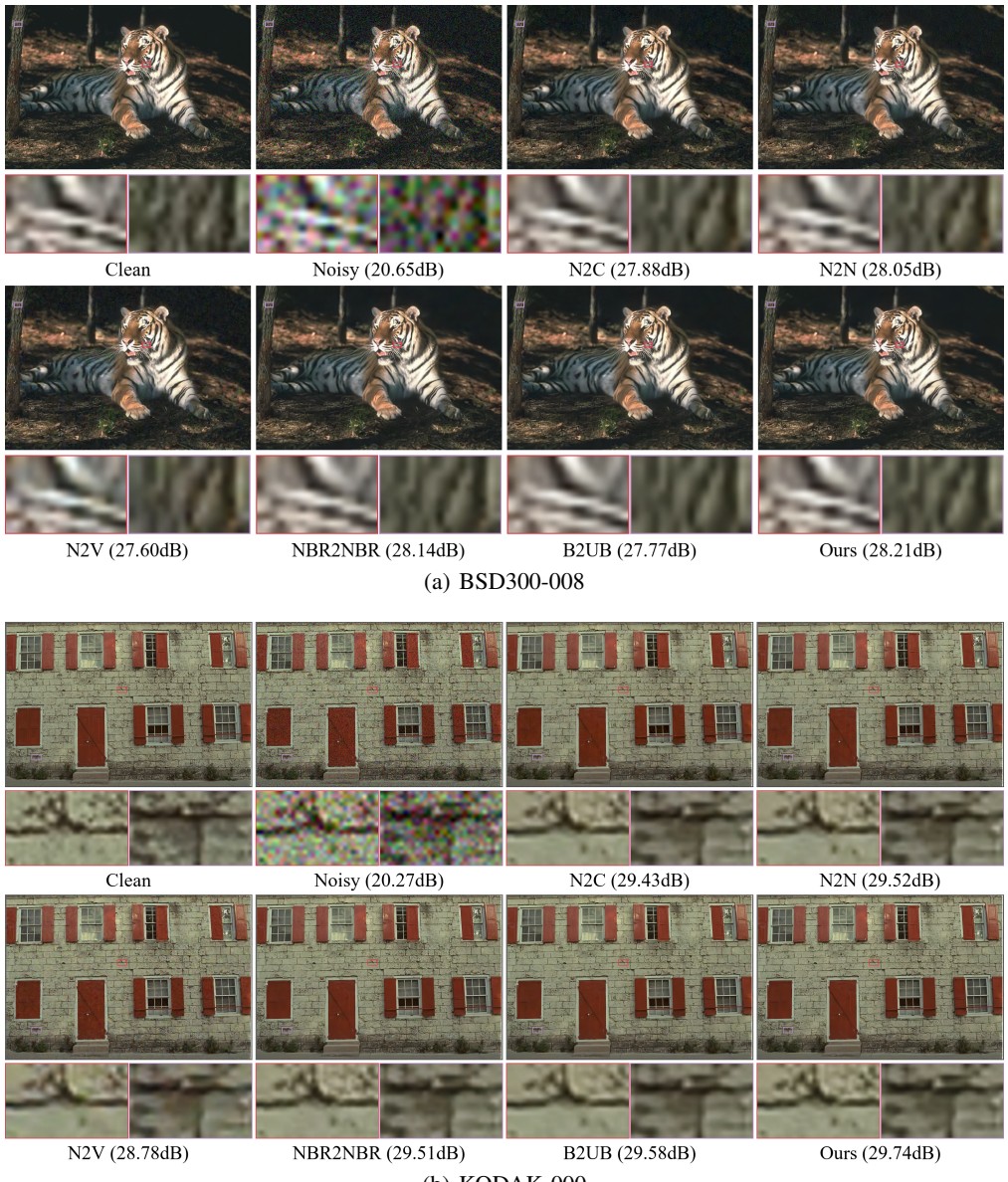

Figure 2: Visual comparison of denoising sRGB images in the setting of $\sigma = 25$.

Table 1: Quantitative denoising results on synthetic datasets in sRGB space. The highest PSNR(dB)/SSIM among unsupervised denoising methods is highlighted in bold.

| Noise Type | Method | KODAK | BSD300 | SET14 |
|---|---|---|---|---|
| Gaussian $\sigma = 25$ | Baseline,N2C Ronneberger et al. (2015) | 32.43/0.884 | 31.05/0.879 | 31.40/0.869 |
| | Baseline,N2N Lehtinen et al. (2018) | 32.41/0.884 | 31.04/0.878 | 31.37/0.868 |
| | CBM3D Dabov et al. (2007) | 31.87/0.868 | 30.48/0.861 | 30.88/0.854 |
| | Self2Self Quan et al. (2020) | 31.28/0.864 | 29.86/0.849 | 30.08/0.839 |
| | N2V Krull et al. (2019) | 30.32/0.821 | 29.34/0.824 | 28.84/0.802 |
| | Laine19-mu Laine et al. (2019) | 30.62/0.840 | 28.62/0.803 | 29.93/0.830 |
| | Laine19-pme Laine et al. (2019) | **32.40/0.883** | 30.99/0.877 | **31.36/0.866** |
| | Noisier2Noise Moran et al. (2020) | 30.70/0.845 | 29.32/0.833 | 29.64/0.832 |
| | DBSN Wu et al. (2020) | 31.64/0.856 | 29.80/0.839 | 30.63/0.846 |
| | NBR2NBR Huang et al. (2021) | 32.08/0.879 | 30.79/0.873 | 31.09/0.864 |
| | B2UB Wang et al. (2022b) | 32.27/0.880 | 30.87/0.872 | 31.27/0.864 |
| | R2R Pang et al. (2021) | 32.25/0.880 | 30.91/0.872 | 31.32/0.865 |
| | Ours | 32.34/0.882 | **31.08/0.879** | 31.20/0.862 |
| Gaussian $\sigma \in [5, 50]$ | Baseline,N2C Ronneberger et al. (2015) | 32.51/0.875 | 31.07/0.866 | 31.41/0.863 |
| | Baseline,N2N Lehtinen et al. (2018) | 32.50/0.875 | 31.07/0.866 | 31.39/0.863 |
| | CBM3D Dabov et al. (2007) | 32.02/0.860 | 30.56/0.847 | 30.94/0.849 |
| | Self2Self Quan et al. (2020) | 31.37/0.860 | 29.87/0.841 | 29.97/0.849 |
| | N2V Krull et al. (2019) | 30.44/0.806 | 29.31/0.801 | 29.01/0.792 |
| | Laine19-mu Laine et al. (2019) | 30.52/0.833 | 28.43/0.794 | 29.71/0.822 |
| | Laine19-pme Laine et al. (2019) | 32.40/0.870 | 30.95/0.861 | **31.21/0.855** |
| | DBSN Wu et al. (2020) | 30.38/0.826 | 28.34/0.788 | 29.49/0.814 |
| | NBR2NBR Huang et al. (2021) | 32.10/0.870 | 30.73/0.861 | 31.05/0.858 |
| | B2UB Wang et al. (2022b) | 32.34/0.872 | 30.86/0.861 | 31.14/0.857 |
| | R2R Pang et al. (2021) | 31.50/0.850 | 30.56/0.855 | 30.84/0.850 |
| | Ours | **32.46/0.875** | **31.13/0.867** | 31.02/0.856 |

Table 2: Quantitative denoising results on SIDD benchmark and validation datasets in raw-RGB space.

| Method | Network | SIDD Benchmark | SIDD Validation |
|---|---|---|---|
| Baseline,N2C Ronneberger et al. (2015) | U-Net | 50.60/0.991 | 51.19/0.991 |
| Baseline,N2N Lehtinen et al. (2018) | U-Net | 50.62/0.991 | 51.21/0.991 |
| BM3D Dabov et al. (2007) | - | 48.60/0.986 | 48.92/0.986 |
| N2V Krull et al. (2019) | U-Net | 48.01/0.983 | 48.55/0.984 |
| Laine19-mu Laine et al. (2019) | U-Net | 49.82/0.989 | 50.44/0.990 |
| Laine19-pme Laine et al. (2019) | U-Net | 42.17/0.935 | 42.87/0.939 |
| DBSN Wu et al. (2020) | DBSN | 49.56/0.987 | 50.13/0.988 |
| NBR2NBR Huang et al. (2021) | U-Net | 50.47/0.990 | 51.06/0.991 |
| R2R Pang et al. (2021) | U-Net | 46.70/0.978 | 47.20/0.980 |
| Ours | U-Net | **50.75/0.991** | **51.29/0.991** |

poor when no pseudo-supervised module participates in training. When $\gamma = 5$, our method achieves the highest accuracy on SIDD.

## 5 CONCLUSION

We propose Generative Recorrupted2Recorrupted, a novel unsupervised denoising framework, which achieves greatly denoising performance without noise prior, and it surpasses methods that require noise prior. The proposed method generates random dynamic noise in the process of training the neural network, so as to solve the problem of requiring noise model prior in unsupervised denoising. In addition, the Pseudo supervised module improve the performance of unsupervised denoising. At last, extensive experiments have shown the superiority of our approach against compared methods.

Table 3: Quantitative denoising results of different $\gamma$ on SIDD validation datasets.

| $\gamma$ | 0 | 2 | 5 | 10 | 15 | 30 |
|---|---|---|---|---|---|---|
| PSNR/SSIM | 49.41/0.989 | 51.23/0.991 | **51.29/0.991** | 51.23/0.991 | 51.20/0.991 | 51.06/0.991 |

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
