# OpenReview forum: "Generative Recorrupted-to-Recorrupted: An Unsupervised Image Denoising Network for Arbitrary Noise Distribution"
_ICLR.cc/2023/Conference — Submitted to ICLR 2023_

### Official Review · Reviewer_c6Mm · 2022-10-16

**Confidence:** 5
**Correctness:** 1
**Technical Novelty And Significance:** 2
**Empirical Novelty And Significance:** 2
**Recommendation:** 3

**Clarity, Quality, Novelty And Reproducibility:**

The paper is quite clear about the main idea. The novelty is very limited and there is severe issues in its theoretical soundness. The reproducibility is reasonable.

**Strength And Weaknesses:**

**Strength**:

1. The paper is quite accessible for catching the main idea.

**Weakness**:
 1. The novelty of the work is limited, it is essentially an extension of R2R with un-justified noise simulation
2. The key theoretical argument for discussing R2R is incorrect.  In the last sentence, the author claim as long as $n+n'$ inside $f$ and $n-n'$ outside out $f$ are independent, the R2R works. It is incrrect, the noise $n+n'$ and $n-n'$ cannot be independent.
3. The key theoretical argument for discussing sign-flipping is wrong. Firstly, the paper claim that the simulated $\hat n$ defined in  (9)  is independent from $n$. It is not true as there is $n$ in the definition of $\hat n$. Secondly, the claim after (1) that $n-\hat n$ and $n+\hat n$ is independent, which is incorrect by the same reason in 2.
4. The performance improvement over existing ones is quite marginal.



**Summary Of The Paper:**

This paper studied the problem of self-supervised image denoising and present an method which is an extension of the existing work R2R (Recorrupted2Recorrupted [Pang et al. CVPR'21).  The basic idea is to synthesize the pair from a noisy image that can simulates the supervised pair (noisy, clean) in supervised learning, where the noise is simulated by randomly flipping the sign of residual. The experiments are conduced in several datasets.

**Summary Of The Review:**

The paper studied an interesting problem, and presents an extension of existing works for solving it. However, the extension is based on some  procedure, and there is severe errors in their theoretical argument. The performance gain is also limited. Thus,, I don't think this paper meets the standard of ICLR for acceptance.

---

### Official Review · Reviewer_L537 · 2022-10-19

**Confidence:** 5
**Correctness:** 1
**Technical Novelty And Significance:** 1
**Empirical Novelty And Significance:** 2
**Recommendation:** 1

**Clarity, Quality, Novelty And Reproducibility:**

Clarity:
1. The proof in Section 3.1:
a) I don't see a detailed theory or proposition corresponding to this proof. The definition of the loss L, the distributions of the noise n, n', and the assumption on the noises, are totally not mentioned. This makes the statement and the proof very vague.
b) What's worse, there are severe errors in the proof. It is said that " as long as n and n' are independent, it can be inferred that n + n' in fθ(\hat{y}) and n − n' are independent". This is totally wrong!
c) It is unclear why Equation (7) can be transformed into Equation (11).

2. The motivation of using sign flipping is rather vague. The paper says the benefit is that sign-flipping generates zero-mean noise. This motivation is rather weak, as there are many other ways to generate zero-mean noise. Since the proof in Section 3.1. I could not get more information what sign-flipping brings to the proposed method. Further, I believe sign-flipping imposes a strong assumption on the noise distribution, but this is not mentioned and empirically studied in the paper either.

---------------------------------------------------------------------------------

Novelty:
The paper is combination of the existing Recorrupted-to-Recorrupted loss and a sign flipping scheme. However, the sign flipping for noise generation has been proposed in [1,2]. The proof related to Recorrupted-to-Recorrupted loss is quoted from the original work with vague explanation, and zero-mean property of the sign-flipping noise generation has been discussed in [1,2]. Therefore, I believe the novelty of the paper is rather limited.

[1] Noise2Grad: Extract Image Noise to Denoise. IJCAI 2021.
[2] Self-Verification in Image Denoising. ArXiv 2021,

--------------------------------------------------------------------------
Quality:
The writing is not good. Many grammar errors and non-standard language usage are found, e.g., the title of Section 3.3.


**Strength And Weaknesses:**

Strengths
1. The proposed method showed improvement over unsupervised methods such as Neighbor-to-Neighbor on the real-world dataset SIDD, though the improvement is somehow marginal.
2. The combination of Recorrupted-to-Recorrupted loss with the sign flipping scheme leads to the improvement.

Weaknesses
1. Novelty is quite limited as the work is the combination of two established works.
2. There are severe errors in the mathematical proof and logic gaps in the explanation of the proposed method.
3. The writing needs improvement. In addition to many grammar errors and non-standard language usage, the technical part is very unclear.

See the comments below for the details about the weaknesses.

**Summary Of The Paper:**

This paper proposed an unsupervised deep learning method for single image denoising, which only takes a set of noisy images for model training. The basic idea is combining the Recorrupted-to-Recorrupted loss with a sign flipping scheme and a pseudo supervised loss on self-generated samples. The experiments on both synthetic Gaussian noise corrupted images and the real-world SIDD dataset showed that the proposed method has certain improvement over the recent unsupervised methods such as Neighbor-to-Neighbor in some settings.

**Summary Of The Review:**

This paper showed the combination of two existing techniques for unsupervised image denoising can lead to certain improvement. However, there are severe errors and many unclear points in the mathematical proof and statements. Such problems are difficult to solve in a revision.

---

### Official Review · Reviewer_CpLn · 2022-10-24

**Confidence:** 4
**Correctness:** 3
**Technical Novelty And Significance:** 3
**Empirical Novelty And Significance:** Not applicable
**Recommendation:** 5

**Clarity, Quality, Novelty And Reproducibility:**

The propose unsupervised framework seems to be effective for synthetic and real noises. Without noise assumption, this work is more applicable than existing methods.

**Strength And Weaknesses:**

Strength: A new unsupervised learning framework for image denoising. The noise prior is not assumed, and the proposed method may be applicable to wider noise types.

Weaknesses:
1. The proposed method is somewhat complicated and is not well explained why it can work. Let us assume the denoiser is perfect, i.e., y-f(y) is exactly the true noise. Then second step does nothing since f can perfectly remove \hat{n}, and third step forces f can remove stronger noise since y+\hat{n} shoulbe be close to y-\hat{n} by L_u. So the main denoising effect may come from the third step. But the contriton of these steps are not validated in ablation study.

2. What is regular loss Lg? And in Fig. 1, network f is w/o gradient and the lines are dashed on Lg. I cannot understand what they mean, and there is no explanations in main context.

3. The English is not inapproriate, e.g., "we will ... perfect denoiser", "PSEUDO SUPERVISED" is not suitable for section title, "the generate noise". Just list a few.

**Summary Of The Paper:**

This paper proposed an unsupervised learning framework for denoising, without assuming noise priors. Three stages are designed, i.e., generating noise, psedo supervised learning and recorrupted2recorrupted. The results on synthetic Gaussian noise and real noise datasets show the proposed method is comparable with existing methods.

**Summary Of The Review:**

The proposed method is effective, but it is not well explained, and the effect of each component is not well verified. The writing needs more effort to polish.

---

### Official Review · Reviewer_XJfe · 2022-10-30

**Confidence:** 4
**Clarity, Quality, Novelty And Reproducibility:** The clarity and quality of this paper…
**Correctness:** 3
**Technical Novelty And Significance:** 2
**Empirical Novelty And Significance:** 2
**Recommendation:** 3

**Strength And Weaknesses:**

Strength:
1. This paper is easy to follow and well-organized.
2. The paper is abundant with experiments. The ablation study somewhat justifies the design choices of the method.

Weaknesses:
1. The main idea of this paper is similar to [1] and [2]. The proposed method seems like a combination of these two methods. I cannot find new technical or theoretical insights in this work.
[1] Lin H, Zhuang Y, Zeng D, et al. Self-Verification in Image Denoising. arXiv preprint arXiv:2111.00666, 2021.
[2] Pang T, Zheng H, Quan Y, et al. Recorrupted-to-recorrupted: unsupervised deep learning for image denoising. CVPR 2021: 2043-2052.
2. The performance improvement is slight according to Fig.2 and Tab.1.

**Summary Of The Paper:**

This paper proposes an unsupervised method for image denoise. The noise model is generated by a residual image and a random mask. With the noise model, the input and target of the network are generated from a single noisy image. Experiments are conducted on both synthetic and real-world datasets.

**Summary Of The Review:**

I think this paper is below the acceptance threshold. The novelty is limited as they just combine several existing ideas.

---

### Decision · Program_Chairs · 2023-01-20

**Decision:**

Reject

**Justification For Why Not Higher Score:**

The paper's contribution is marginal as argued by the reviewers, the theoretical argument appears to contain errors, and the paper is not ready for publication. All reviewers recommend to reject the paper, and I agree with their recommendation.

**Justification For Why Not Lower Score:**

N/A

**Metareview: Summary, Strengths And Weaknesses:**

The paper proposes an un-supervised method for image denoising. The method does not require clean imageas, and does not make assumptions on the noise model.

Strength: The paper considers a relevant problem, i.e., unsupervised image denoising, and attempts to provide both theoretical and empirical results.

Weaknesses: The paper's contribution is marginal as argued by the reviewers, the theoretical argument appears to contain errors, and the paper is not well written, not ready for publication.

**Summary Of Ac-Reviewer Meeting:**

N/A